# Don't Add, don't Miss: Effective Content Preserving Generation from Pre-Selected Text Spans

**Aviv Slobodkin[1], Avi Caciularu[1,2], Eran Hirsch[1], Ido Dagan[1]**

[1]Bar-Ilan University   [2]Google Research
{lovodkin93, hirsch.eran}@gmail.com
avica@google.com
dagan@cs.biu.ac.il

## Abstract

The recently introduced Controlled Text Reduction (CTR) task isolates the text generation step within typical summarization-style tasks. It does so by challenging models to generate coherent text conforming to pre-selected content within the input text ("highlights"). This framing enables increased modularity in summarization-like tasks, allowing to couple a single CTR model with various content-selection setups and modules. However, there are currently no reliable CTR models, while the performance of the existing baseline for the task is mediocre, falling short of practical utility. Here, we address this gap by introducing a high-quality, open-source CTR model that tackles two prior key limitations: inadequate enforcement of the content-preservation constraint, and suboptimal silver training data. Addressing these, we amplify the content-preservation constraint in both training, via RL, and inference, via a controlled decoding strategy. Further, we substantially improve the silver training data quality via GPT-4 distillation. Overall, pairing the distilled dataset with the highlight-adherence strategies yields marked gains over the current baseline, of up to 30 ROUGE-L points, providing a reliable CTR model for downstream use.[1]

## 1 Introduction

The abstractive text summarization task, aiming to generate accurate and coherent summaries from one or multiple documents, involves two principal sub-tasks: (a) identification of salient information in the input text(s) and (b) its consolidation into a coherent text. Recently, Slobodkin et al. (2022) proposed an explicit decomposition of these two subtasks, particularly concentrating on the latter as an isolated task termed *Controlled Text Reduction* (CTR), as illustrated in Figure 2. This task takes as input a text with pre-selected marked spans ("highlights") and expects a reduced and coherent version of the text, covering precisely the content of these input spans. In addition to a baseline model, the authors also provided crowdsourced dev and test sets, along with automatically-generated silver training data, where each instance comprises of a document with highlighted content and the corresponding reduced text summary. The proposed adoption of CTR offers greater control over text generation, enabling modular summarization systems, where a single CTR model can be combined with various content selection strategies and user preferences. For example, the same CTR model could be used for generation from content selected for either generic or query-focused summarization (QFS; Dang, 2006), or for long-form question-answering (LFQA; Fan et al., 2019). It could even enable a human-in-the-loop scenario, where customized summaries can be created, based on users' preferences, as was recently demonstrated in Slobodkin et al. (2023b). Further, by excluding the *subjective* content selection requirement of the full summarization task, CTR offers a semantically well-defined and objective generation task, focused on coherent content consolidation.

Being recently introduced, there currently exists no high-quality CTR model, with the present baseline succeeding to cover only half of the highlighted details while pulling much non-highlighted content from the surrounding context. In this paper, we aim to design an efficient CTR model of sufficient quality for reliable integration in modular architectures. Our proposed method, outlined in Figure 1, addresses two different shortcomings of the existing resources. First, we examine methods to intensify the highlights signal during training and inference, in order to yield better content preservation. Second, we address the noise evident in the available silver training data, improving its quality using GPT-4.

---

[1]Our code is publicly available at https://github.com/lovodkin93/CDR_CTR

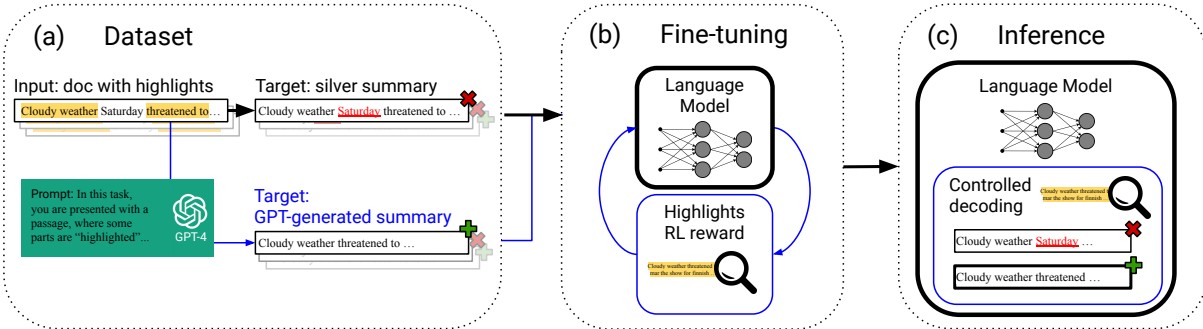

Figure 1: Overview of our contributions, encompassing three modeling phases. Components introduced in our approach are denoted in blue. (a) We generate new target summaries using GPT-4, conditioned on the silver highlights in the original dataset. (b) During training, we fine-tune our model taking an RL approach, based on Quark (Lu et al., 2022a). (c) During inference, we employ a highlights-centric controlled decoding algorithm.

We begin our investigation by exploring strategies for enforcing the highlights signal. First, we explore the use of Reinforcement Learning (RL) during *training* to both accentuate the highlights signal and mitigate the impact of the data noise, adapting the recently-introduced Quark algorithm (Lu et al., 2022a), which aims to *un*learn unwanted properties. Since RL methods bias the model to conform to a reward function, in addition to the training data, we hypothesize it has the potential to reduce the impact of noisy data biases on the model's behavior. We then design a highlight-aware decoding mechanism, following Wan et al. (2023), which biases the model to better preserve the highlights content at *inference* time. Finally, we address the inherent noise within the available silver training data, by employing GPT-4 (OpenAI, 2023) to generate cleaner training data for our model, essentially performing (symbolic) distillation from GPT-4.

Empirically, we demonstrate that each of the aforementioned strategies separately yields state-of-the-art results, surpassing the baseline model in terms of highlights content preservation. Further, we show that GPT-4 is indeed effective in generating better silver training data, leading to further improvements.

Hence, our contribution in this paper is twofold:

1. Proposing and investigating multiple strategies to amplify the highlights signal in the CTR setting, addressing training, inference, and data generation.

2. Developing a high-quality CTR model, significantly outperforming the available baseline.

## 2 Background

This section provides the needed background regarding our task and the methods we employ.

**Controlled Text Reduction** Controlled Text Reduction (CTR; Slobodkin et al., 2022) is a recently introduced task that aims to generate a reduced version of a text that *exactly* covers pre-determined selected content, referred to as "highlights" (see Figure 2). It effectively generalizes the sentence decontextualization task (Choi et al., 2021), which addresses only the case of rephrasing a single full sentence given in context to be comprehensible standalone. In contrast to the full summarization task, which involves a substantial degree of subjectivity in content selection, CTR requires exact preservation of modularly *pre-selected* content, making the task more semantically objective. At the same time, this stringent requirement for both faithfulness to and coverage of the highlights, makes the task more semantically challenging than standard summarization, which is obliged only to faithfulness, but not to full coverage.

Accompanying the task, the CTR authors introduced a manually annotated development and test sets, as well as an automatically-generated silver train set, derived from the DUC summarization dataset[2] using a summary-source alignment model (SuperPAL; Ernst et al., 2021). However, an approximate 30% mismatch surfaced between the 'silver highlights', namely source spans identified by SuperPAL as related to the summary, and their respective summary content. Such discrepancy may cause a model trained on this data to develop biases toward overlooking certain types of highlights

---

[2]https://duc.nist.gov/

Figure 2: Demonstration of the Controlled Text Reduction task. The input consists of a source document and highlights (left), and the desirable output covers exclusively the highlighted content while preserving coherence (right). Borrowed and adapted from Slobodkin et al. (2022).

during training, as well as including salient, yet non-highlighted content. Here, we address this issue by improving the CTR training dataset.

**Controlling via Reinforcement Learning** Reinforcement Learning (RL) has been increasingly utilized to control various facets of text generation (Pasunuru and Bansal, 2018; Yuan et al., 2019; Nakano et al., 2021), with most works relying either on REINFORCE (Williams, 1992), an algorithm notable for its direct yet high-variance approach to policy optimization, or the Proximal Policy Optimization (PPO; Schulman et al., 2017), an algorithm renowned for efficiently balancing policy stability and learning. There has also been a growing interest in using RL methods for reducing undesired behaviors, including toxicity (Faal et al., 2022) and redundancy (Mao et al., 2020).

Following this line of work, Lu et al. (2022a) recently introduced Quark - an algorithm inspired by the Proximal Policy Optimization algorithm, designed to *un*learn unwanted properties, which we leverage here. While REINFORCE and PPO are used to learn a desired policy, Quark aims to remove or reduce specific behaviors from the learned policy, enabling it to effectively address the undesired behaviors in the text generation process. The algorithm iteratively alternates between three steps: (1) **Exploration**, where the current model state generates new input-output samples from the training inputs, and then incorporates them into the stored data pool; (2) **Quantization**, where a predefined reward function is used to rank and classify the accumulated data pool into $K$ quantiles, with a reward token assigned to each; and (3) **Learning**, where the algorithm maximizes the likelihood of the accumulated data pool, with the instances conditioned on their quantile labels. A KL-divergence penalty is also applied during training to ensure proximity to the original language model distribu-

tion (Jaques et al., 2016; Ziegler et al., 2019).

The objective of Quark is to teach the model to generate texts of varying quality with respect to the reward function. Then, at inference, the model is asked to generate high-reward outputs. The algorithm exhibits state-of-the-art results in several attribute-removal objectives, such as toxicity and unwanted sentiment, surpassing many other RL-based text generation approaches, including PPO.

In this work, we propose modifying Quark to teach models to unlearn biases caused by the somewhat noisy training data, thereby enhancing their performance in adhering to the highlighted content.

**Controlled Decoding** Controlled decoding algorithms aim to guide models to better address concrete output requirements that can be measured and enforced during decoding. To this end, various works designed constraint-sensitive decoding methods that modify the search space in accordance with the constraints (Anderson et al., 2017; Hokamp and Liu, 2017; Post and Vilar, 2018; Lu et al., 2021; Slobodkin et al., 2023a).

Recently, a faithfulness-aware decoding mechanism was proposed by Wan et al. (2023), which involves a lookahead operation during decoding, inspired by Lu et al. (2022b). At every decoding step, the algorithm projects into the future to create a complete summary commencing with the current tokens of any partially formed summary. It then selects tokens that provide paths exhibiting enhanced faithfulness within the search space. Formally, each token's score is calculated by:

$$f(y_t) = logP(y_{\leq t}|x) + \lambda \cdot \max_{\mathcal{L}_l(y_{\leq t})} g(y_{\leq t+l}, x) \quad (1)$$

where $x$ represents the current input tokens, $t$ is the generation step, $y$ is the generated output, $logP(y_{\leq t}|x)$ is the underlying generation score, $g(\cdot)$ is a faithfulness evaluation function, $\lambda$ stands

as a hyperparameter to regulate the emphasis placed on future predictions, and $l$ stands for the number of tokens to look into the future. $\mathcal{L}_l(y_{\leq t})$ is a collection of length $l$ continuations of $y_{\leq t}$, and its size ranges from a single continuation in a greedy decoding approach, to $k$ potential continuations in beam-search decoding. In this work, we adapt this algorithm, shifting its focus towards precise matching with the highlighted content rather than being (only) faithful to the entire input.

**Data Generation with Language Models**  Recently, many works proposed using LM-generated data to directly train smaller manageable models. These efforts address various challenges, including controllability (Sclar et al., 2022), model reasoning (Zelikman et al., 2022; Hsieh et al., 2023), and language understanding (Ye et al., 2022; Han et al., 2022). These works align with the Symbolic Knowledge Distillation scheme (West et al., 2022), where knowledge from the teacher model is transferred via a textual dataset used for student training. Here, we employ GPT-4 to generate improved silver training data for our models.

## 3   Method

This section presents the three core components of our contribution: highlights-driven reinforcement learning (RL) (§3.1), highlight-attentive controlled decoding (§3.2), and the steps taken for generating improved fine-tuning data with GPT-4 (§3.3), corresponding to components (b), (c), and (a) in Figure 1, respectively.

### 3.1   Highlights-Oriented RL Training

To adapt Quark (see §2) for the CTR task, we introduce a highlights-focused reward, based on the ROUGE metric (Lin, 2004).[3] Previous RL-based text generation studies with ROUGE-based rewards calculated ROUGE scores relative to the gold reference summaries. In contrast, we propose calculating the ROUGE rewards for the generated output compared to the concatenated *input highlights*, to encourage their content preservation. Furthermore, to motivate the model to balance between the task's two requirements, namely, covering the entire highlights and avoiding the inclusion of excessive non-highlighted content, we suggest optimizing each of these objectives separately. Inspired by the dual-reward procedure (Pasunuru and Bansal, 2018),

---

[3]We also experimented with PPO, but adopted Quark which outperformed it on the development set.

we propose alternating between two highlights-focused rewards: One that encourages coverage of highlights, for which we use ROUGE *recall*, and another that prioritizes adherence (faithfulness) to the highlights, for which we employ ROUGE *precision*. Indeed, we find that this alternating reward strategy works best over the development set (see Appendix A).

### 3.2   Highlights-Sensitive Decoding

To bias the model to better preserve the highlights content at inference time, we follow the faithfulness-aware decoding strategy from Wan et al. (2023) (see §2). Here, we adapt this method to prioritize partially formed summaries that are likely to eventually (once completed) match better the pre-selected highlights. To that end, we substitute the score $g(y_{\leq t+l}, x)$ in Equation 1 with a new score, $g(y_{\leq t+l}, x_h)$, where $x_h$ represents the concatenation of the highlights. In essence, our strategy shifts from an input-focused to a highlights-focused scoring approach, while requiring both faithfulness and complete coverage. Within this framework, we evaluate two potential metrics for the computation of $g(y_{\leq t+l}, x_h)$: ROUGE-L F1 and METEOR scores. We found that ROUGE-L F1 consistently matched or exceeded METEOR's performance, and hence adopted it for our method. Please refer to Appendix B for further details.

### 3.3   Improving Silver Training Data with GPT-4

We wish to improve the quality of the CTR dataset, by leveraging the capabilities of GPT-4 (OpenAI, 2023). The existing CTR training dataset consists of summaries manually composed by experienced summarizers, while the highlights were automatically identified using a summary-source alignment model (see §2). As discussed, the performance of this alignment model leaves much room for improvement.

To capitalize on GPT-4's abilities in text generation, we employ it to generate more fitting summaries, based on the silver highlights. To that end, we supply GPT-4 with a modular prompt, inspired by the Chain-of-Thought approach (Wei et al., 2023). This custom prompt incorporates two exemplars that deconstruct the controlled reduction into three steps: (1) the listing of highlights, (2) the consolidation of highlights on a sentence-by-sentence basis, and (3) the production of the

ultimate reduction. The detailed structure of this prompt can be found in Appendix C.

We will henceforth refer to models trained on the GPT-4-generated data as distilled, and to those trained on the original CTR data as non-distilled, following the notion of Symbolic Knowledge Distillation (West et al., 2022). As shown in Section §5.3, we observed an improved quality of the dataset, yielding better alignments between highlights and summaries.

## 4 Experimental Setup

**Base Model**   Throughout our experiments, our primary base model is the instruction-finetuned Flan-T5 large model (Flan-T5$_{large}$; Chung et al., 2022), further fine-tuned on the highlights-focused CTR dataset. The selection of this particular model is motivated by emergent research that indicates instruction-finetuned models manifest superior performance in tasks necessitating constrained generation (Sanh et al., 2022; Wei et al., 2022; Zhou et al., 2023). We will refer to this model as Flan-T5$_H$, where H stands for "highlight-finetuned".

In addition to this variant of Flan-T5$_H$, we also show results of a large variant of the original pre-trained CTR baseline model, which is the Long-former Encoder-Decoder large model (LED$_{large}$; Beltagy et al., 2020), finetuned on the CTR dataset. We will refer to this model as LED$_H$. Lastly, we also show the results of the few-shot GPT-4 (OpenAI, 2023), when guided with the same prompt used in our distillation process (see §3.3).

**Highlights-Oriented RL Training**   Our highlights-sensitive RL training involves fine-tuning the anchor Flan-T5$_H$ model using Quark, combined with our highlights-driven reward policy (see §3.1). Specifically, the first Exploration step (see §2) utilizes the pretrained and fine-tuned Flan-T5$_H$ to generate the initial pool of samples, which triggers the Quark unlearning process, performed on Flan-T5$_H$. Following Lu et al. (2022a), we set the number of quantiles to eight (see §2). We find, on the development set, that ROUGE-L is the most effective metric for our dual-reward function, yielding the greatest improvements relative to the ROUGE-1 and ROUGE-2 metrics (henceforth, ROUGE rewards refer to ROUGE-L).

**Highlights-Sensitive Decoding**   To score the degree of alignment between each future completion of a current generated prefix and the highlights, we

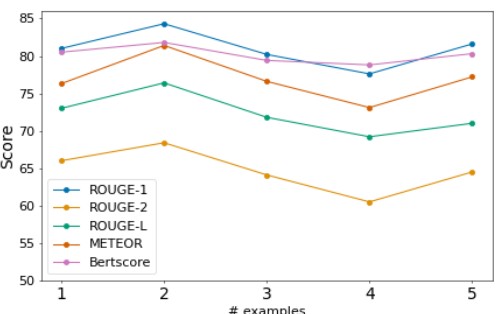

Figure 3: ROUGE, METEOR, and BertScore results on 50 instances from the CTR development set of GPT-4 models, for varying numbers of in-context examples in the prompt.

utilize the F1 measure of ROUGE-L, relative to the concatenated highlights (see §2 and §3.2). We use the hyperparameters used in Wan et al. (2023). Additionally, following its tuning, we chose a beam size of 8. For a more comprehensive discussion of the hyperparameter tuning, see Appendix B.

**Generating New Training Data with GPT-4**   As part of our methodology to improve the generation of the silver training data, we evaluated the effect of varying the number of examples used in the few-shot prompt for GPT-4. This procedure involved the random sampling of fifty instances from the CTR development set, and their subsequent incorporation within a few-shot prompt. The quantity of in-context examples was systematically varied between one and five, the results of which are documented in Figure 3. These results show that the use of two examples delivered the most favorable results across all evaluative metrics, thereby leading us to select this configuration in our experiments. We also experimented with ChatGPT and found it to be inferior to GPT-4, leading to the adoption of GPT-4 in this setting. For a more in-depth discussion of these findings and additional hyperparameter tuning, please refer to Appendix D.

**Evaluation**   For evaluation, we adopt the evaluation approach outlined by Slobodkin et al. (2022), calculating several content-matching metrics between the generated texts and the concatenated highlights. The rationale behind evaluating against the highlights, rather than the gold summaries, is rooted in CTR's principal requirement: an exact matching to the *highlighted content*, not necessarily to the reference summary. While the highlights and reference summary are supposed to express the

| | model | R-1 | R-2 | R-L | M | BertScore | Coherency |
|---|---|---|---|---|---|---|---|
| Non-Distilled | $LED_H$ (baseline) | 70.2 | 53.8 | 54.4 | 64.3 | 71.9 | 4.26 |
| | $Flan$-$T5_H$ | 74.1 | 59.1 | 63.4 | 67.4 | 73.2 | 4.56 |
| | + RL | 75.6 | 63.1 | 68.9 | 73.6 | 76.8 | 4.34 |
| | + Con. Decoding | **80.7** | **68.6** | **76.1** | 74.2 | **80.0** | 4.40 |
| | + RL + Con. Decoding | 78.1 | 66.4 | 73.4 | **75.1** | 76.8 | 4.42 |
| Distilled | $Flan$-$T5_H$ | 84.3 | 72.4 | 79.9 | 80.9 | 75.8 | 4.32 |
| | + RL | 85.0 | 74.3 | 82.1 | **84.3**[*] | 81.9 | 4.40 |
| | + Con. Decoding | **88.3**[*] | **79.1**[*] | **86.4**[*] | 84.0 | **84.2**[*] | 4.54 |
| | + RL + Con. Decoding | 86.7 | 76.4 | 84.6 | 84.1 | 81.9 | 4.44 |
| | GPT-4 | 82.3 | 67.1 | 75.1 | 79.2 | 80.8 | 4.58 |

Table 1: ROUGE, METEOR (M), and BertScore results on the CTR testset, compared to the concatenated highlights, as well as coherency results. In addition to the baseline $LED_H$ and $Flan$-$T5_H$, we also evaluate the combination of $Flan$-$T5_H$ with the highlights-sensitive decoding strategy ("+ Con. Decoding"), with the highlights-focused RL strategy ("+ RL"), and with their combination. We also evaluate all those variants when trained with the GPT-4-generated trainset ("Distilled") as well as GPT-4 itself, in a few-shot setting with two exemplars. For each metric, the best non-distilled and distilled Flan-T5 models are in bold, with the best overall model having an asterisk.

same content, some discrepancies may occur even in the gold test data. Moreover, since automatic metrics are not perfect, models that are less abstract than the human-generated gold summaries, or that exhibit different paraphrastic abstractions, would unjustly be penalized when evaluated against the gold summaries. In Appendix E, we conduct a qualitative analysis that further shows the advantage of directly comparing outputs with the highlights, rather than with the gold summaries.

In addition to ROUGE (Lin, 2004), used in Slobodkin et al. (2022), we also measure METEOR scores (Denkowski and Lavie, 2014), informed by recent research underscoring the correlation of this metric with human judgment concerning relevance and consistency (Fabbri et al., 2021). We also report results on the BertScore metric, which is less lexical and more semantic in nature, though not necessarily more reliable. Experimenting also with NLI-based metrics, we observed inadequate performance when applied over sub-sentence spans (our highlights), and hence leave it for future research to develop NLI-based metrics that are robust to such settings.

Finally, we also follow Slobodkin et al. (2022)'s coherency analysis, by hiring crowd-workers to assess the coherency of 50 random samples for each model, with a 5-point Likert scale (for details see Appendix F). Notably, our coherency analysis tested both coherency and fluency, following the settings in the original CTR paper. This approach follows the standard common practice, where fluency and coherence are best evaluated manually.

## 5 Results and Analysis

### 5.1 None-Distilled Models

Table 1 presents performance results, for models trained over the original CTR silver training data (Slobodkin et al., 2022)). Primarily, $Flan$-$T5_H$ exhibits superior performance compared to $LED_H$. This trend is consistently apparent in all subsequent variants, hence, we confine our reporting of subsequent models to $Flan$-$T5_H$'s variants.[4]

We observe that further finetuning $Flan$-$T5_H$ via our highlights-oriented RL protocol ("+ RL" in Table 1) yields substantial improvements. Additionally, augmenting $Flan$-$T5_H$ with the highlights-aware decoding strategy ("+ Con. Decoding" in Table 1) leads to an even larger performance improvement across ROUGE and BertScore metrics and a modest improvement in the METEOR metric. We find this sensible, given the aggressive nature of the controlled decoding approach in amplifying the highlight-signal when actually generating the output at inference time. Interestingly, the incorporation of both strategies simultaneously results in a slight drop in performance based on the ROUGE and BertScore metrics compared to the controlled decoding variant, yet it yields the optimal outcomes on the METEOR metric. Critically, all our model variants maintain a human fluency score above 4.3 (out of 5), demonstrating that the fluency requirement is not compromised. This is not surprising, given the impressive ability of current language

---

[4]See Appendix G for the results of all the variants with $LED_H$ as the backbone model.

models to generate fluent texts.

## 5.2 Distilled Models

From Table 1 we observe a noteworthy finding: when trained on the GPT-4-generated data, Flan-T5$_H$ surpasses all the non-distilled alternatives, which were trained on the original CTR data. Notably, it also surpasses the GPT-4 model, where the latter is prompted in the few-shot setting, just as it was when generating the training data for Flan-T5$_H$. Given the inherent limitation of GPT-4 in that it is not open-sourced and cannot be fine-tuned on our data, this puts it at a distinct disadvantage compared to the other models, thereby obstructing its broader application.

We also note that Flan-T5$_H$ appears to further benefit from both the auxiliary highlights-focused RL finetuning and the incorporation of the highlights-attentive decoding strategy. We find that the decoding strategy outperforms the RL approach based on ROUGE and BertScore metrics, similar to the findings for the non-distilled models, while the RL approach appears to be more effective on the METEOR metric. We also note that the combination of both strategies does not result in any additional advantages. Ultimately, just as in the non-distilled setting, we find that the coherency of the generated outputs remains uncompromised. In total, the combination of the highlights-focused strategies with the improved GPT-4-generated data leads to significant improvements, outperforming the baseline model by over 30 ROUGE-L points, and resulting in a reliable and effective CTR model for future downstream use.

## 5.3 Distillation Data Quality Assessment

To shed light on the distillation success, we analyze the quality of the generated dataset. To that end, we first sample 10 GPT-4-generated instances and manually identify their corresponding highlights, which we treat as "gold" highlights. Then, we apply Slobodkin et al. (2022)'s automated silver annotation methodology (see §2) on the GPT-4-generated summaries, leading to a new set of highlights for each pair of input text and GPT-4-generated summary. These highlights, combined with their corresponding inputs and summaries, represent the original CTR dataset. Alternatively, the original highlights presented to GPT-4, combined with the input texts and the summaries it generated, represent the new dataset. Subsequently, we proceed to calculate the ROUGE-L F1 score between

|  | Faithfulness (P) | Coverage (R) | F-1 |
|---|---|---|---|
| LED$_H$ | 65.8 | 72.9 | 69.2 |
| Flan-T5$_H$ | 71.1 | 74.0 | 72.5 |
| Flan-T5$_H$ (distil.) | 79.1 | 90.8 | 84.6 |
| + RL | 81.3 | **93.4** | 86.9 |
| + Con. Decoding | **85.6** | 91.3 | **88.3** |
| + RL + Con. Decoding | 83.4 | 92.3 | 87.7 |

Table 2: Fact-wise faithfulness (P), coverage (R), and F-1 scores between generated summaries and the highlighted spans for LED$_H$, Flan-T5$_H$, and four variants of the distilled Flan-T5$_H$: regular, with the RL finetuning ("+RL"), with the controlled decoding approach ("+ Con. Decoding") and with their combination. For each metric, the best models are in bold.

the manually annotated highlights and each of the automatically-generated variants. Our analyses reveal that the GPT-4-generated instances demonstrate a significantly greater alignment with the manually-annotated instances than those from the original dataset, achieving a ROUGE-L score of 79%, as opposed to 67.9%.

## 5.4 Manual Performance Analysis

To further evaluate the effectiveness of various components within our approach, we adopt the manual analysis methodology proposed in the CTR paper (Slobodkin et al., 2022). Consistent with the original authors' procedure, we select 10 random samples from the test set. Subsequently, we compute precision, recall, and F-1 scores for highlighted information units for a selection of six models utilized in this study: LED$_H$, Flan-T5$_H$, the distilled Flan-T5$_H$, and the three variants of the distilled Flan-T5$_H$ equipped only with RL training, only with the controlled decoding, and with both approaches, respectively.[5] Our calculations cover 195 highlighted input units and approximately 180 system summary units for each model. The results of this analysis are illustrated in Table 2.

We observe that Flan-T5$_H$ exhibits significantly increased faithfulness to the highlights compared to LED$_H$, with a modest improvement in highlight coverage. Conversely, training on the GPT-4-generated data markedly augments adherence to the highlights as well as leading to significant highlight coverage, thus demonstrating the approach's inherent value.

Our findings also suggest that the application of RL finetuning primarily improves highlight coverage. The seemingly modest contribution to faith-

---

[5]For further details, refer to Appendix H.

fulness could potentially be attributed to the RL mechanism's attempt to achieve a balance between reward optimization and coherence-preservation (due to the KL-divergence term), a requirement that might necessitate incorporating additional information from the surrounding context. Alternatively, the deployment of highlight-centric controlled decoding displays a more pronounced impact on adherence to highlights. This phenomenon could potentially be attributed to the strategy's more rigorous enforcement of highlight constraints, thereby enforcing limited deviation from the highlights. The modest advancement in coverage could be attributed to the lookahead mechanism of controlled decoding. At every generation step, the algorithm favors tokens whose 'greedy' completion leads to better highlight adherence, rather than exploring multiple potential trajectories for each candidate akin to a k-beam style. In doing so, this method inadvertently neglects additional, more suited candidates, whom a beam search could capture and thereby enhance coverage. Lastly, the combination of both strategies results in a mid-point performance between the individual gains attained by each strategy in both faithfulness and coverage. These findings indicate that while merging the strategies does enhance the weaker model's performance in each metric, it also has a trade-off effect on the other metric. Future research will explore methods to harness the complementary aspects of these strategies while mitigating this trade-off effect.

## 6 Discussion and Future Work

Our empirical assessments reveal the merits of each of the proposed methods. Specifically, the utility of the distillation process emerges as complementary to both reinforcement learning (RL) training and controlled decoding. The latter methods still have room to play in both enforcing the content preservation constraints beyond the sheer training data, as well as in overcoming a certain level of noise that persists also in the GPT-4 generated data (as shown in §5.3).

Conversely, the combination of controlled decoding and highlight-centric RL approach did not yield improvements over the better-performing method (controlled decoding, in our experiments). Given these strategies impact different stages of the generation process, namely training and inference, they may possess the potential for synergy. For example,

the decoding strategy may be integrated into the RL training's sampling phase, possibly increasing the model's awareness of the decoding constraints during training. Our performance analysis, detailed in §5.4, further supports this hypothesis by demonstrating that each technique amplifies different aspects of the highlights: one notably improves their coverage while the other augments adherence, suggesting the exploration of this potential synergy in future research.

Furthermore, the existing implementation of highlight-centric controlled decoding is computationally demanding, given its requirement to generate a complete completion for every candidate token at each generation step. Wan et al. (2023) motivated the need to generate entire summaries by the expectation of current faithfulness metrics for full summary inputs. Yet, as our method does not employ faithfulness metrics, it would be insightful to explore how to leverage partial summaries, rather than complete ones, and thus to significantly reduce the computational overhead.

Finally, given the high-quality highlight adherence and coverage exhibited by our best models, investigating their incorporation into modular summarization pipelines emerges as a promising research direction. We suggest exploring this direction in future research.

## 7 Conclusion

In this study, we addressed the lack of a high-quality Controlled Text Reduction (CTR) model by focusing on two pivotal aspects: the amplification of the highlight signals and the mitigation of noise within the training data. We started by proposing two distinct strategies aiming at augmenting the highlights signal. The first strategy emphasized this signal during training, where we combined an RL approach with a custom highlights-oriented reward. The second strategy was introduced during inference, where we employed a controlled decoding mechanism that prioritizes generation paths ensuring higher adherence to the highlights. Furthermore, we addressed the intrinsic noise in the CTR dataset by generating new instances using GPT-4, significantly enhancing the dataset quality.

Empirical evidence shows the effectiveness of our proposed methodology. Each of our highlight-centric strategies individually led to significant improvements over the baseline model in terms of highlight-matching capabilities. Additionally,

training on the GPT-4-generated data yielded further improvements, outperforming each of the non-distilled variants, trained on the original CTR dataset. In total, our highest-performing models achieved state-of-the-art results, outperforming the baseline by more than 30 ROUGE-L points, while also preserving comparable levels of coherency. Future work would focus on improving the combination and efficiency of the different components, as well as on the incorporation of our best-performing models in modular summarization pipelines.

## 8 Limitations

Although the risk involved in our work is minimal, like other advanced language generation models available today, we cannot ensure that our model will consistently produce accurate information, despite our attempts to utilize clean highlighted data and controlled decoding. Hence, it is crucial to exercise caution when employing the model in real-world scenarios and thoroughly test it before deploying it.

In addition, the controlled decoding method we use is limited by its slow speed, which may hinder its practicality for production systems. Additionally, the rigid constraints imposed during decoding can restrict the model's linguistic flexibility and creative potential, potentially limiting its suitability for generating varied or innovative outputs. These considerations highlight the need to carefully evaluate the trade-offs before implementing controlled decoding in production environments, and hence should be further studied in future research.

## 9 Ethics Statement

This paper is submitted in the wake of a tragic terrorist attack perpetrated by Hamas, which has left our nation profoundly devastated. On October 7, 2023, thousands of Palestinian terrorists infiltrated the Israeli border, launching a brutal assault on 22 Israeli villages. They methodically moved from home to home brutally torturing and murdering more than a thousand innocent lives, spanning from infants to the elderly. In addition to this horrifying loss of life, hundreds of civilians were abducted and taken to Gaza. The families of these abductees have been left in agonizing uncertainty, as no information, not even the status of their loved ones, has been disclosed by Hamas.

The heinous acts committed during this attack, which include acts such as shootings, sexual assaults, burnings, and beheadings, are beyond any justification.

In addition to the loss we suffered as a nation and as human beings due to this violence, many of us feel abandoned and betrayed by members of our research community who did not reach out and were even reluctant to publicly acknowledge the inhumanity and total immorality of these acts.

We fervently call for the immediate release of all those who have been taken hostage and urge the academic community to unite in condemnation of these unspeakable atrocities committed by Hamas, who claim to be acting in the name of the Palestinian people. We call all to join us in advocating for the prompt and safe return of the abductees, as we stand together in the pursuit of justice and peace.

## Acknowledgements

This work was supported by the Israel Science Foundation (grant no. 2827/21), and a grant from the Israel Ministry of Science and Technology.

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

## A  Reward Tuning

We train the modified dual-reward Quark model with three pairs of highlights-oriented ROUGE rewards: ROUGE Precision+ROUGE F1, ROUGE Recall+ROUGE F1, and ROUGE Precision + ROUGE Recall. From Table 3, which shows the ROUGE scores on the CTR development set, it arises that alternating between ROUGE precision and ROUGE recall rewards yields the best results.

| R-L Rewards | R-1 | R-2 | R-L | M | BertScore |
|---|---|---|---|---|---|
| Precision + F1 | 74.0 | 60.7 | 67.3 | 69.8 | 76.3 |
| Recall + F1 | 74.1 | 60.9 | 67.6 | 70.4 | 76.2 |
| Only F1 | 74.4 | 61.2 | 67.3 | 70.0 | 76.4 |
| Precision + Recall | **76.7** | **64.0** | **70.4** | **74.0** | **77.7** |

Table 3: ROUGE, METEOR (M), and BertScore results of the Flan-T5$_H$ model, combined with the RL strategy, on the CTR development set. We experimented with different pairs of alternating highlights-oriented ROUGE-L rewards, as well as only a single ROUGE-L F1 reward.

| | R-1 | R-2 | R-L | M | BertScore |
|---|---|---|---|---|---|
| METEOR | 77.0 | 60.3 | 61.7 | **71.3** | 74.3 |
| ROUGE-L | **78.3** | **63.9** | **72.3** | **71.3** | **76.4** |

Table 4: F1 results for ROUGE, METEOR (M), and BertScore compared to the concatenation of the highlights on the dev set. The results are for Flan-T5$_H$ with a beam size of 2, incorporating two variants of the highlights-centric score during decoding: METEOR (F1) compared to the highlights, and ROUGE-L (F1) compared to the highlights. **Bold** marks the highest results on each metric.

## B  Highlights-Focus Controlled Decoding Hyperparameter Tuning

For our highlights-oriented decoding strategy, we experiment with two scores: METEOR score and ROUGE-L F1 score. These are juxtaposed against the concatenated highlights to assess the potential completion's adherence to the highlights. Table 4 shows the results on the CTR development set. We note that the ROUGE-L score consistently offers superior or at least equivalent performance in comparison to the METEOR score. Therefore, based on these findings, we make the informed decision to adopt the ROUGE-L score as our primary evaluation metric in all subsequent experiments. Furthermore, we undertake a series of experiments with varied beam sizes encompassing 2, 4, 6, and 8. The outcomes, as observed on the development set, are illustrated in Figure 4. Corroborating the findings of Wan et al. (2023), we discern a marginal impact of the increase in beam size on regular decoding. However, its influence on the highlight-focused controlled decoding escalates in parallel with the beam size across all metrics. In light of these findings, we opt for a beam size of 8 for the remainder of our experiments.

## C  GPT-4 Prompt

For deploying GPT-4, we use a prompt that consists of some basic instructions, as well as two

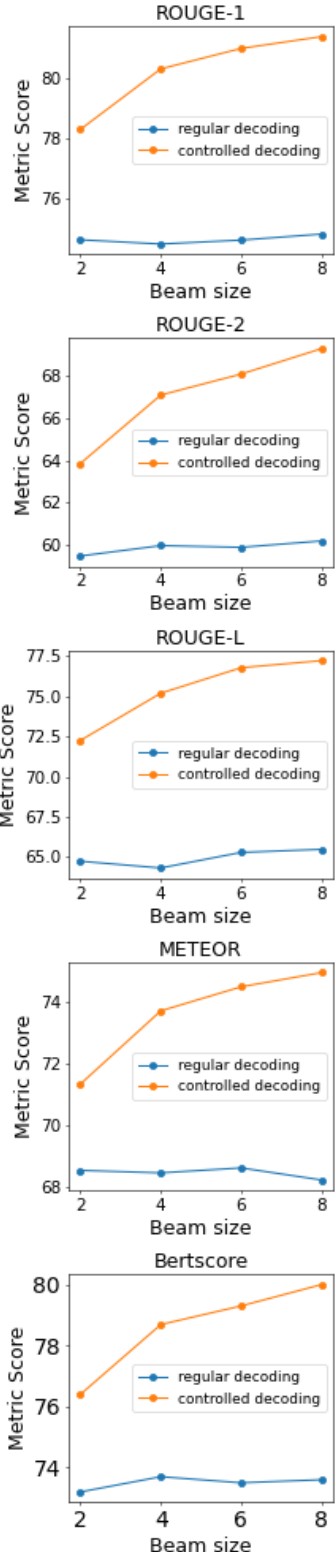

Figure 4: The scores of fine-tuned Flan-T5$_H$ with controlled decoding compared to regular decoding at various beam sizes. We present a comparison of the generated summary to the highlights concatenation.

exemplars. The in-context examples consist of a modular generation, where we separate the task

| Model | | R-1 | R-2 | R-L | M | BertScore |
|---|---|---|---|---|---|---|
| GPT-3.5 | regular | 59.9 | 40.7 | 47.7 | 49.7 | 70.0 |
| | modular | 72.7 | 61.1 | 68.1 | 75.0 | 75.7 |
| GPT-4 | regular | 71.0 | 55.1 | 64.5 | 63.1 | 76.4 |
| | modular | **81.0** | **66.0** | **73.0** | **76.3** | **80.5** |

Table 5: ROUGE, METEOR (M), and BertScore results on 50 instances from the CTR development set of GPT-3.5 and GPT-4 models. For each model, we prefixed an exemplar to the prompt, in two fashions: "regular", where the exemplar simply generated the custom summary; and "modular", where the exemplar first performed the intermediate sentence-by-sentence highlights consolidation, before generating the final summary.

| | R-1 | R-2 | R-L | M | BertScore |
|---|---|---|---|---|---|
| w\o list$_h$ | 84.3 | 68.4 | 76.4 | 81.4 | **81.8** |
| with list$_h$ | **84.8** | **71.3** | **77.1** | **83.6** | 81.8 |

Table 6: ROUGE, METEOR and BertScore results, compared to the concatenated highlights, on 50 instances from the CTR development set of GPT-4 models, once when the prompt does not contain highlights enumeration of the current instance ("w\o list$_h$") and once when it does contain it ("with list$_h$").

into three steps: (1) highlights extraction and enumeration, (2) highlights consolidation sentence-by-sentence, and (3) generation of the final reduction (see Figure 6). Additionally, we already perform the highlights extraction step for the current example, leaving only the consolidation and generation of the final reduction to the model.

## D Prompt-Tuning

To improve the quality of the CTR trainset, we test both GPT-4 and GPT-3.5 (ChatGPT), both with modular prompting and with regular prompting, where each exemplar's answer consists solely of the final reduction. From Table 5, which shows the ROUGE and METEOR scores on 50 instances of the CTR development set, we observe that indeed the modular approach substantially improves both models' performances and that GPT-4 is superior to ChatGPT in handling the task. Additionally, from a manual analysis of GPT-4's generations, we find that occasionally it misses a highlighted span in the highlights-listing step, leading to its absence from the final summary. Consequently, we also experiment with a prompt that already incorporates the highlights-listing step, leaving only their consolidation and the generation of the final reduction to the model. This version indeed which improves the model's performance (see Table 6), making it the final version we use.

## E Qualitative Analysis

Figure 7 exhibits two instances from the test set, consisting of input texts along with their corresponding highlights. Accompanying each instance are the outputs generated by the non-distilled Flan-T5$_H$, the output generated by the distilled Flan-T5$_H$, and the original gold summaries.

In the first example, the non-distilled model achieved a ROUGE-L score of 58.6 when compared to the gold summary, while its distilled counterpart received a score of 46.5. Despite the apparent superiority of the non-distilled model, we note that while it did cover all the highlighted spans, it also added several non-highlighted segments ("an Atlanta-based cable network", "the next business of the scientist", "the network's" and "look for" in the context of looking for cause and effect). In contrast, the distilled model, besides capturing all highlighted content, only added "look for". Consequently, the distilled model delivered a more fitting output, contradicting its supposed deficiency in terms of the ROUGE-L metric relative to the gold summary. Alternatively, when compared to the concatenated highlights, which yielded scores of 68.8 for the non-distilled model, and 78.9 for the distilled model, the results are more reflective of the actual performance.

In the second example, the distilled model amounts to 83.3 in comparison to the concatenated highlights, as opposed to a mere 43.2 when compared to the gold summary. Interestingly, despite the fact that its output nearly covers all the highlighted spans (except for "with a chance of showers for...Finland" and "in Helsinki" in the context of where the eclipse would end), and introduces very minimal non-highlighted information ("thousands of", "3,000" and "the national airline"), it still results in a lower ROUGE-L score when aligned with the gold summary. This discrepancy can be attributed to two significant factors. Firstly, the distinction in abstraction techniques employed between the gold summary and the generated output can contribute to the reduced scores. Secondly, the gold summary introduces information that is absent from the original input text, and therefore from the highlights ("on July 20"). Both these elements can explain the divergence between the model's performance as per the ROUGE-L metric

and its actual competency. This discrepancy can be resolved when directly comparing the generated summary to the highlights, which offers a more accurate reflection of its quality.

## F  Fluency Human Annotation Protocol

We ask crowd-workers to rate the fluency of the texts generated by the baseline supervised model, the PPO model and the dual-reward Quark model. Our group of crowd-workers consists of reliable workers that have shown a good understanding of different semantic tasks including summarization in previous experiments. To evaluate, we randomly select 100 documents from our test set and evaluate their corresponding generated text by the three aforementioned models (300 samples in total). We design a simple Amazon Mechanical Turk interface, where we present each time one of the 300 samples (see Figure 5). Following Slobodkin et al. (2022) we use a 5-point Likert scale to evaluate the fluency of the generated summaries. Additionally, we add criteria explaining each score, to reduce ambiguity and ensure consistent ratings (see Figure 5). Assessing an average response period of 30 seconds, we priced each response with 10 ¢.

## G  Results of All Variants with LED$_H$ as the Backbone Model

Table 7 shows performance results for all the variants explored in this work, with LED$_H$ as the backbone model.

## H  Performance Analysis Settings

To further examine the efficiency of our different components, we follow Slobodkin et al. (2022) and manually assess several of our models on two levels: (1) precision to the highlighted content and (2) recall of the highlighted spans. For that, we compare each system summary span to the source highlights. To that end, we randomly select 10 samples from our test set, with their corresponding system summaries (one for each of the models - 50 in total). Then, following the notion of Summary Content Unit (SCU) in the Pyramid method for summarization evaluation (Nenkova and Passonneau, 2004), we extract such units from both the summary and the source highlighted spans using the Summary Evaluation Environment (SEE) interface, described in their paper.

Then, to calculate the precision, for each summary unit, we manually search for a matched high-lighted unit conveying the same information, to determine whether the summary unit is mentioned in the highlights (TP) or not (FP), and then calculate the (micro-)precision to the highlighted content.

For the recall calculations, we also count the number of False Negative (FN) summary facts, compared to the facts in the highlights, namely highlighted information units that were absent from the system summaries. Then, combined with the TP count, we calculate the (micro-)recall of the highlighted content.

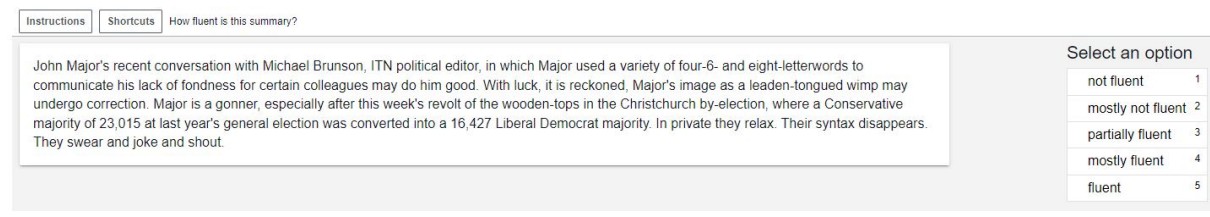

**Instructions**

In this task, you are asked to rate the quality of a passage.
To correctly solve this task, please follow the following steps:

1. Read the passage.
2. Rate it on a scale from **1** (worst) to **5** (best) by its fluency.

**Definition of Fluency**

This rating measures the quality of the text - are the sentences well written and grammatically correct, do they fit together and sound natural.
Consider how legible, grammatical and coherent the passage is.
The scale should be:

- **1** - The summary is incoherent, with multiple grammatical errors and not natural-sounding.
- **2** - The summary contains only small parts that are coherent, with many grammatical errors and sentences that fit together poorly.
- **3** - The summary is somewhat coherent, though it either contains several grammatical errors or sentences that don't fit very well.
- **4** - The summary is mostly coherent, with little to no grammatical errors and sounds natural enough.
- **5** - The summary is very coherent, without any grammatical errors and sounds natural.

| Instructions | Shortcuts | How fluent is this summary? |

John Major's recent conversation with Michael Brunson, ITN political editor, in which Major used a variety of four-6- and eight-letterwords to communicate his lack of fondness for certain colleagues may do him good. With luck, it is reckoned, Major's image as a leaden-tongued wimp may undergo correction. Major is a gonner, especially after this week's revolt of the wooden-tops in the Christchurch by-election, where a Conservative majority of 23,015 at last year's general election was converted into a 16,427 Liberal Democrat majority. In private they relax. Their syntax disappears. They swear and joke and shout.

Select an option
| | |
|---|---|
| not fluent | 1 |
| mostly not fluent | 2 |
| partially fluent | 3 |
| mostly fluent | 4 |
| fluent | 5 |

Figure 5: Example of the data collection interface used by the crowd-workers to evaluate the fluency of summaries.

| | model | R-1 | R-2 | R-L | M | Bertscore |
|---|---|---|---|---|---|---|
| Non-Distilled | LED$_H$ | 70.2 | 53.8 | 54.4 | 64.3 | 71.9 |
| | + RL | 70.8 | 56.1 | 56.5 | **70.9** | 73.7 |
| | + Con. Decoding | **77.4** | **64.7** | **71.9** | 69.5 | **77.8** |
| | + RL + Con. Decoding | 75.4 | 60.9 | 67.5 | 68.6 | 76.7 |
| Distilled | LED$_H$ | 77.9 | 64.0 | 69.7 | 75.6 | 74.0 |
| | + RL | 76.6 | 61.7 | 66.8 | 75.4 | **78.2**[*] |
| | + Con. Decoding | **81.4**[*] | **68.5**[*] | **77.9**[*] | **75.5**[*] | 77.1 |
| | + RL + Con. Decoding | 78.1 | 63.8 | 73.0 | 73.1 | 75.6 |

Table 7: ROUGE, METEOR, and Bertscore results on the CTR testset, compared to the concatenated highlights, of all the different variants tested in this work, with LED$_H$ as the backbone model. In addition to the baseline LED$_H$, we also evaluate the combination of LED$_H$ with the highlights-sensitive decoding strategy ("+ Con. Decoding"), with the highlights-focused RL strategy ("+ RL"), and with their combination. We also evaluate all those variants when trained with the GPT-4-distilled trainset ("Distilled"). For each metric, The best non-distilled and distilled LED$_H$ models are in bold, with the best overall model having an asterisk.

```
 1   In this task, you are presented with a passage, where some parts are "highlighted" (namely,
         there are <highlight_start and <highlight_end> tokens before and after each such span).
         Your job is to generate a summary that covers all and only the "highlighted" spans.
 2
 3   Example1:
 4   Passage: <highlight_start>Ben Johnson spent his homecoming<highlight_end> in seclusion,
         <highlight_start>without the<highlight_end> Olympic <highlight_start>gold medal
         and<highlight_end> the <highlight_start>hero's welcome<highlight_end>, as Canadians
         bemoaned the fate of the sprinter who failed the drug test.
 5   ...
 6
 7   Answer: The highlighted spans are:
 8    1. Ben Johnson spent his homecoming
 9   ...
10   The highlights spans are combined as follows:
11   Spans 1,2,3,4 are combined to form sentence 1: Ben Johnson spent his homecoming without
         the gold medal and hero's welcome.
12   ...
13   So, the answer is:
14   Ben Johnson spent his homecoming without the gold medal and hero's welcome.
15   ...
16
17   Example2:
18   Passage: Japanese writer <highlight_start>Kazuo Ishiguro won the<highlight_end> 1989
         <highlight_start>Booker Prize<highlight_end>, Britain's top literary award,
         <highlight_start>for his novel "The Remains of the Day<highlight_end>," judges
         <highlight_start>announced Thursday<highlight_end>.
19   ...
20
21   Answer: The highlighted spans are:
22    1. Kazuo Ishiguro won the
23   ...
24   The highlights spans are combined as follows:
25   Spans 1,2,3,4 are combined to form sentence 1: It was announced Thursday that Kazuo
         Ishiguro won the Booker Prize for his novel "The Remains of The Day".
26   ...
27   So, the answer is:
28   It was announced Thursday that Kazuo Ishiguro won the Booker Prize for his novel "The
         Remains of The Day".
29   ...
30
31   Now your turn:
32   Passage:  <highlight_start>The motion picture industry's most coveted
         award<highlight_end>, <highlight_start>Oscar<highlight_end>, <highlight_start>was
         created 60 years ago and 1,816 of the statuettes have been produced so
         far.<highlight_end>
33   ...
34   Answer: The highlighted spans are:
35    1. The motion picture industry's most coveted award
36   ...
37   The highlights spans are combined as follows:
```

Figure 6: Example prompt provided to GPT-4. The prompt consists of basic instructions, two in-context examples, and the instance input. The examples demonstrate a modular pipeline, where we first extract the highlights, then consolidate them sentence-by-sentence, and lastly generate the final reduction.

**Input:** Weather experts discussed the drought Thursday and its possible causes and effects, and said they hoped to produce a consensus on how much longer it will last. All of the speakers at the 1988 Drought Symposium called for more research and study. "There's an enormous number of possibilities that one could chase. The next business of the scientist is to eliminate the possibilities of... and extend the range of analysis and look for cause and effect," said David Rodenhuis, director of the federal Climate Analysis Center in Washington. The symposium attended by 13 state and national experts was sponsored by The Weather Channel, an Atlanta-based cable network, which broadcast the meeting live. John Hope, the network's hurricane specialist and a former forecaster with the National Hurricane Center in Miami, Fla., said the drought in the Southeast might be lessened or ended in the next few months by a heavier than normal hurricane season. Hope said current weather conditions point to more tropical storms, more of which should become hurricanes, in this hurricane season. He stressed that a hurricane's effect on drought is variable, depending on the intensity and speed of the storm. Nevertheless, he said, some Southeastern states get as much as 25 pecent of their rainfall in some months from hurricanes. Georgia state climatologist Gayther Plummer told the symposium he thought the position of the planet Jupiter had something to do with drought in the Southeast. However, Rodenhuis said he was skeptical about Plummer's thesis. "It's very necessary to be cautious and analytical about the issue," he said.

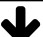

**Not distilled :** The 1988 Drought Symposium was sponsored by The Weather Channel, an Atlanta-based cable network. 13 state and national experts attended. David Rodenhuis, director of the federal Climate Analysis Center in Washington, said the next business of the scientist is to eliminate possibilities and extend the range of analysis and look for cause and effect. John Hope, the network's hurricane specialist, said that the drought in the Southeast might be lessened or ended by a heavier than normal hurricane season. Some Southeastern states get as much as 25% of their rainfall in some months from hurricanes. Georgia state climatologist Gayther Plummer said he thought the position of Jupiter had something to do with the drought, but Rodenhuis was skeptical.

**Distilled :** The 1988 Drought Symposium was held to eliminate the possibilities of the drought and extend the range of analysis to look for cause and effect, according to David Rodenhuis, director of the federal Climate Analysis Center in Washington. The symposium, attended by 13 state and national experts, was sponsored by The Weather Channel. John Hope, a hurricane specialist, said that the drought in the Southeast might be lessened or ended by a heavier than normal hurricane season. Some Southeastern states get as much as 25% of their rainfall in some months from hurricanes. Georgia state climatologist Gayther Plummer, who spoke at the symposium, thought the position of the planet Jupiter had something to do with the drought, but Rodenchides was skeptical about his thesis.

**Gold :** Today's 1988 Drought Symposium, sponsored by The Weather Channel, was attended by 13 state and national Weather experts. David Rodenhuis, director of the federal Climate Analysis Center in Washington, stressed the need to eliminate cause-and-effect possibilities and extend the range of analysis. Georgia state climatologist Gayther Plummer thought the drought in the SE was related to the position of Jupiter, but Rodenhuis remained skeptical. Hurricane specialist John Hope suggested the SE drought might be lessened or ended by a heavier than normal hurricane season. Some SE states get as much as 25% of their rainfall some months from hurricanes.

- - - - - - - - - - - - - - - - - - - - - - - - - - - - - - - - - - - - - - -

**Input:** Cloudy weather Saturday threatened to mar the show for thousands of Finnish and foreign skygazers hoping to glimpse a total solar eclipse in this land of the midnight sun. The weather forecast took a turn for the worse in the evening, when the Finnish Meteorological Services predicted cloudy weather with a chance of showers for eastern Finland on Sunday. In the eastern town of Joensuu, a television news broadcast late Saturday showed it was already cloudy there with a light drizzle falling. The solar eclipse in Finland starts at 4:03 a.m. Sunday (9:03 p.m. EDT Saturday). At that time, the moon will begin gradually moving between the Earth and the sun. The total eclipse begins at 4:52 a.m. in Helsinki and will last 83 seconds. After the total phase of the eclipse, the moon will move away, uncovering more and more of the sun. The eclipse ends at 5:45 a.m. in Helsinki. About 10,000 people _ including 3,000 foreigners _ have converged on Joensuu, about 50 miles from the Soviet border. There conditions there are considered especially good for viewing the eclipse _ weather permitting. In Helsinki, the total eclipse phase will occur 16 minutes after sunrise, when the sun is only 1 degree above the horizon. At Joensuu, 310 miles northeast of Helsinki, the sun will be 5 degrees above the horizon in the total phase at a better angle for watchers. The sun rises unusually early during summer in the extreme northern latitudes where Finland is located. The sun comes up unusually late during winter. Ten months ago, Joensuu hired an "eclipse secretary" to handle arrangements for the expected influx of visitors. But that official, Marjut Cadia, said she had underestimated the interest in the event. "We completely sold out the 10,000 special eyeglasses we made for this event, and our extra stock is finished too," she said in a television interview Saturday. During the past week, newspapers, television and radio have been full of information about solar eclipses, as well as advice for spectators not to stare directly into the sun. Some eclipse viewers won't have to worry about the clouds, because they will be above them. Finnair, the national airline, has arranged a dozen special flights for eclipse watchers, and private companies with small planes will provide more. The eclipse will be total in an arc about 125 miles wide from the northern Baltic Sea and southeast Finland, across the Kola Peninsula and northeast Soviet Union, to the Aleutian Islands near Alaska. Scientists will conduct several experiments during the eclipse, including gravity test measurements of the radius of the sun. However, scientific interest in this eclipse has been less than in longer eclipses, such as the seven-minute eclipse seen from northern Kenya on June 30, 1973. Seppo Linnaluoto of the Ursa Astronomical Association said the best place for observations of this eclipse will be the northeast Soviet Union. The last total eclipse to be seen in Finland occurred in 1945.

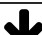

**Not distilled :** About 10,000 people, including 3,000 foreigners, have converged on Joensuu, 50 miles from the Soviet border, to view a total solar eclipse. Conditions there are considered especially good for viewing. The total eclipse will begin at 4:52 a.m. and last 83 seconds. The last total eclipse to be seen in Finland occurred in 1945. The eclipse will be total in an arc about 125 miles wide. The clouds will be above the viewers. Finnair, the national airline, has arranged special flights for eclipse watchers.

**Distilled :** Cloudy weather threatened to mar the show for thousands of Finnish and foreign skygazers hoping to glimpse a total solar eclipse in Finland. The total eclipse begins at 4:52 a.m. in Helsinki and will last 83 seconds. The eclipse ends at 5:45 a,m. In Joensuu, about 10,000 people, including 3,000 foreigners, have converged on the town, with conditions considered especially good for viewing the eclipse. Some eclipse viewers won't have to worry about the clouds, as they will be above them. Finnair, the national airline, has arranged special flights for eclipse watchers, with the eclipse being total in an arc about 125 miles wide. The last total eclipse to be seen in Finland occurred in 1945.

**Gold :** Cloudy, rainy weather threatened to ruin viewing of a total solar eclipse in Finland on July 20. Best viewing, weather permitting would be in the eastern town of Joensuu and about 10,000 people, including foreigners have gathered there. The eclipse begins at 4:52 am in Helsinki and ends at 5:45 am with the total phase lasting for 83 seconds. Finnair has arranged flights for some viewers to go above the weather to view the eclipse. The eclipse will be total in an a 125 mile wide arc in northern arctic areas. The last total eclipse viewed in Finland was in 1945.

Figure 7: Examples including the input, gold output and predictions with various baselines.