# OpenReview forum: "Don’t Add, don’t Miss: Effective Content Preserving Generation from Pre-Selected Text Spans"
_EMNLP/2023/Conference — EMNLP 2023 Findings_

### Official Review · Reviewer_vhDS · 2023-07-21

**Soundness:** 4

**Excitement:**

4: Strong: This paper deepens the understanding of some phenomenon or lowers the barriers to an existing research direction.

**Paper Topic And Main Contributions:**

The paper makes progress on the controlled text reduction task by using a better signal from the training data (highlights) during the training (reinforcement learning) and inference (constrained decoding).

**Questions For The Authors:**

1. Ad L267: Why is it important to alternate between rewards and not simply use a convex combination of $\lambda r_1 + (1-\lambda) r_2$?

**Reasons To Accept:**

The proposed combination of method adjustements (RL+decoding) seems to be very potent at the task and work well. This could become a widely adapted approach for controlled text reduction.

**Reasons To Reject:**

The paper was very difficult for me to parse, which can be attributed either to the writing organization or, more likely, my lack of expertise in this area.

**Reproducibility:**

2: Would be hard pressed to reproduce the results. The contribution depends on data that are simply not available outside the author's institution or consortium; not enough details are provided.

**Reviewer Confidence:**

1: Not my area, or paper was hard for me to understand. My evaluation is just an educated guess.

**Typos Grammar Style And Presentation Improvements:**

- According to title capitalization rules, even the second _"don't"_ in the title should be capitalized.
- Figures 1 and 7 are included as raster images. They appear to be generated using matplotlib or some derivative thereof. You can use `plt.savefig("figname.pdf")` before `plt.show()` to get the PDF (vector) version which you can include. This way the image will be of higher quality.
- L215, L218: Use `\log` instead of `log`.

---

> ### Author Rebuttal · Authors · 2023-08-28
>
> We would like to thank the reviewer for the appreciation of our work.
>
> ***Main Concern - difficulty to follow the paper***
>
> Thank you for bringing up this issue in a sincere manner. It is possible that the paper was organized for an audience familiar with our area. We will carefully check this aspect and provide additional background information and overall roadmaps to make the paper more easily accessible for broader readership.
>
> ***Question***
>
> In preliminary stages of our research, we did in fact experiment also with convex combinations of the two rewards in certain settings, and found them suboptimal compared to the alternating reward approach (which is also prominent in NLG for handling multiple rewards). Hence, we proceeded with the alternating reward scheme. Following your question, we performed additional experiments with convex combinations on our current models and settings, and found that these consistently lead to a decrease in performance across all metrics. We will add these results to the appendix, and will refer to them in the paper.

---

### Official Review · Reviewer_j7hm · 2023-08-04

**Typos Grammar Style And Presentation Improvements:** N/A
**Soundness:** 3

**Excitement:**

2: Mediocre: This paper makes marginal contributions (vs non-contemporaneous work), so I would rather not see it in the conference.

**Missing References:**

N/A

**Paper Topic And Main Contributions:**

This paper focuses on the controlled text reduction task and processes a new CTR model.  The CTR model consists of two strategies: Highlights-Oriented RL Training and Highlights-Sensitive Decoding, and also  improve the quality of the CTR dataset by using GPT-4.

**Questions For The Authors:**

Question A: Why not test the two proposed strategies/GPT-4 distill data on the baseline's backbone model？
Question B: See Reasons To Reject1

**Reasons To Accept:**

1. Well-written paper. The paper is easy to follow.
2. Extensive experimentation is conducted.

**Reasons To Reject:**

1.  The contribution of the whole paper is limited.  The strategy of the proposed CTR model is almost derived from existing work, which greatly reduces the contribution of this paper. Even more baffling to me is the fact that experiments have shown that the two strategies (RL and Controlled Decoding) mentioned in the method decrease performance when used together, and the authors do not analyze this phenomenon in more depth. This makes me completely confused as to what the purpose of the proposed RL strategy is.
2. The experimental part was grossly inadequate. The author only conducted experiments on Flan-T5, and the backbone of baseline compared with Flan-T5 was not Flan-T5 (it can be seen in Table 1 that the original FLAN-T5 had better performance than baseline). It is difficult to explain whether the effectiveness of this method is due to the selection of a strong backbone. At the same time, there are serious defects in the human evaluation experiment. The author did not evaluate the CTG model with two strategies for human evaluation. This is a very important point, which can rule out whether the lower score of the combination of the two strategies in the automatic evaluation is because the evaluation metrics are not good enough.

**Reproducibility:**

3: Could reproduce the results with some difficulty. The settings of parameters are underspecified or subjectively determined; the training/evaluation data are not widely available.

**Reviewer Confidence:**

3: Pretty sure, but there's a chance I missed something. Although I have a good feel for this area in general, I did not carefully check the paper's details, e.g., the math, experimental design, or novelty.

---

> ### Author Rebuttal · Authors · 2023-08-28
>
> We would like to thank the reviewer for the insightful comments, and would now address them one-by-one.
>
> ***Issue with the limited contribution:***
>
> In this paper, our main goal was developing a reliable controlled reduction model for downstream applications. This was motivated by the appealing potential applicability of this task in providing modularity and increased control in text generation, contrasted with the fact that the only available baseline model for the task had limited performance (~0.5 in both coverage and faithfulness). Since the main challenge in this task is to adhere to very tight constraints with respect to the input (covering strictly the highlighted content, faithfully and fully, while pulling only decontextualization information from the context), we opted to comprehensively explore relevant approaches for biasing models to adhere to the task constraints. This led us to cover the three approaches presented in the paper: symbolic knowledge distillation, RL and controlled decoding. While these general approaches are known, we had to tackle non-trivial challenges in order to apply them in the controlled reduction setting. In particular, the distillation phase required developing a high-performing chain-of-thought scheme, while the RL method required investigating different reward schemes, leading to a dual reward that addresses the two primary constraints of the task (coverage and faithfulness to highlights). While successful application of controlled decoding required less effort, altogether we had to address substantial challenges to come up with a high performing system (80’s-90’s), now enabling downstream usages.
>
> In summary, while our paper is not focused on a particular algorithmic novelty, we view it as providing the following contributions: 1) comprehensive investigation of relevant strategies, including negative conclusions, for driving the CTR model to obey the task constraints; 2) non-trivial technical solutions, primarily regarding distillation and RL; and 3) a high-performing tool for the task for downstream use, thus achieving our targeted goal.
>
> ***Issues regarding the combination of the two approaches (RL and controlled decoding):***
>
> We thank the reviewer for raising the issues regarding method combination, in the second parts of comments 1 & 2. We now realize it deserves more attention. Driven by your comment, we performed the following, to be added to the paper:
>
> 1) (per the 2nd part of issue #2) We conducted human evaluation also for the combination of the methods over our best performing backbone model (Flan T5 Distilled). The results are presented below, along with those of each method separately (as in the paper). We also add the F1 score for each line, to allow also a single-score comparison:
> | Algorithm           |    Faithfulness (P)    |     Coverage (R)    | F1|
> |-------------------|:----------------------:|:--------------------:|:--------:|
> | + RL                |           81.3         |          93.4       |   86.9   |
> | + Con. Decoding     |           85.6         |          91.3       |   88.3   |
> | Combined            |           83.4         |          92.3       |   87.7   |
>
> As seen, these results are quite consistent with the corresponding behavior in the automatic evaluation (Table 1, which pertains to F1 scores). Overall, while the RL approach yields higher Recall and Con. Decoding yields higher Precision, the combined strategy yields some “midpoint” between the standalone methods, with a more balanced Recall-Precision tradeoff. Overall, in terms of F1 scores over the evaluated samples, the combination improves a little over the RL approach while performing slightly worse than the controlled decoding only, which is consistent with most automatic metrics as well.
>
> 2) (per the 2nd part of issue #1) In addition, we performed a qualitative analysis, over the manually evaluated data, of the mismatching classifications between the combined model and each model alone. Notably, we found that for the RL method, the majority of False Positives, that is, content appearing in the summary which is not highlighted in the source, corresponds to non-highlighted spans within densely highlighted source contexts, where the highlighted and non-highlighted spans were syntactically related. In these False Positives, when generating a summary sentence, the RL method failed to exclude the non-highlighted content, which would have required some rephrasing. For instance, in the following example, the RL-based model failed to exclude the non-highlighted content, and extractively outputted the full source sentence (**bold** marks highlights): “**The new surveys led USDA experts to predict the 1988 corn crop will total** 4.48 billion bushels, **down** 37 percent **from last year's harvest** and the smallest output since 1983”. \
> &nbsp; &nbsp;
> In some of these cases, the incorporation of controlled decoding, which checks for highlighting adherence at inference time, succeeded in filtering out False Positive content. Yet, occasionally it omitted also the adjacent highlighted content, thus reducing coverage. This behavior points at the need to develop better controlled decoding algorithms than the one we leveraged in this work, which would be more successful in reranking beams that continuously preserve highlighted content while excluding non-highlighted information. Such an algorithm may better leverage the higher-quality beams generated by the RL model. Developing such a novel controlled decoding algorithm for inference time, which successfully couples with RL training, remains a challenge for future research. \
> &nbsp; &nbsp;
> Overall, our paper provides a first thorough investigation of appealing approaches for imposing the strict biases of the controlled decoding task. Our findings indicate that each of RL, controlled decoding, and their combination currently yields a different faithfulness-coverage tradeoff balance, while calling for future work to develop a systematically superior combined approach.
>
> ***Claim of insufficient experimentation with the backbone model***
>
> (1st part of issue #2 and Question A)
> If properly understood, we suspect that this comment might have arised due to some confusion or misses regarding certain information that does appear in the paper. We point at this information below, and will make sure in the final version to make it more prominent and augment it with details expected by the reviewer.
>
> As mentioned in lines 435-441 (specifically 439-441), in our experiments, we compared the addition of our three contributions (distillation, RL, controlled decoding) over both the baseline model (LED) and our primary model (Flan T5), as backbones, testing all variants. As stated in these lines, LED was consistently inferior as backbone, in each configuration, relative to Flan T5, Consequently, and as stated, we included in Table 1 only the performance of the LED as the bare backbone model (1st line in the table), pointing at it’s inferiority relative to the bare Flan T5 (the second line), as we considered the figures of its consistently inferior performance in all configurations to be of lesser importance. Following the reviewer's comment, we will add this detailed information to the paper (possibly in an appendix), which will also show that the impact of our three contributions and their combinations over this model is consistent with their impact on Flan T5 (as was originally found in our experiments, but not mentioned in the paper).

---

### Official Review · Reviewer_AKMy · 2023-08-05

**Typos Grammar Style And Presentation Improvements:** It lacks some description and statist…
**Soundness:** 4

**Excitement:**

4: Strong: This paper deepens the understanding of some phenomenon or lowers the barriers to an existing research direction.

**Paper Topic And Main Contributions:**

As a subtask of abstractive text summarization, controlled text reduction (CTR) is a recently introduced task which aims to generate a reduced and coherent text for a text with pre-selected content (i.e., highlights). This paper then proposes an approach by introducing three components. First, it uses GPT-4 to generate new summaries of the training data. Second, it fine-tunes the pre-trained model (i.e., the instruction-finetuned Flan-T5 large) with an existing RL but focuses on the source-side pre-selected context. Third, in inference it uses an existing controlled decoding algorithm but again focuses on the source-side pre-selected context. Experimental results demonstrate the effectiveness of the proposed approach.

**Questions For The Authors:**

Is there any way to evaluate the quality of original training data and the GPT-4 generated training data?

**Reasons To Accept:**

1) The proposed approach with three components is well motivated and reasonable.
2) The approach achieves significant improvement over baselines.

**Reasons To Reject:**

1) Both the second and third components are borrowed from relevant studies with only slight changes.

**Reproducibility:**

3: Could reproduce the results with some difficulty. The settings of parameters are underspecified or subjectively determined; the training/evaluation data are not widely available.

**Reviewer Confidence:**

3: Pretty sure, but there's a chance I missed something. Although I have a good feel for this area in general, I did not carefully check the paper's details, e.g., the math, experimental design, or novelty.

---

> ### Author Rebuttal · Authors · 2023-08-28
>
> We would like to thank the reviewer for appreciating our work.
>
> ***Components borrowed for prior work***
>
> In this study, we sought to develop a high-performing CTR model, given the appeal of the task for downstream use vs. the mediocre performance of the available baseline model. Our methodology was to comprehensively investigate appealing paradigms for biasing the generation model to comply with the relatively strict constraints of the CTR task. Developing a suitable RL method for the CTR setup has actually been quite involved, eventually following a non-standard Reinforced Unlearning approach (QUARK) and modifying it with a dual-reward training scheme that addresses the two primary CTR constraints (coverage and faithfulness wrt the highlights). Indeed, the controlled decoding algorithm we adopted required only introducing a suitable reranking score. Analyzing its results, though, suggests interesting directions for future work on developing a better controlled decoding algorithm, deserving a dedicated research project. Overall, the methods we developed indeed achieve our original goal, yielding a high-performing CTR model, while setting the grounds for further research.
>
> ***Question - training data quality***
>
> We did in fact assess the quality of both training datasets in section 5.3 (lines 497-519), and showed that the GPT-4 generated data is of higher-quality.

---

### Meta-Review · Area_Chair_S7E2 · 2023-09-19

**Recommendation:** 3

**Metareview:**

This paper studies the controlled text reduction task. The reviewers find the paper to be well-motivated, and attempt a combination of reasonable strategies. The paper also seems to benefit from good writing. However, the reviewers find the novelty of the work to be extremely limited with most of the techniques borrowed from the literature. The experiment section needs more work, with thorough human evaluation and clearly identifying the source of improvement. Given the importance of the task, this paper could establish baselines for the text reduction task in the future.

---

### Decision · Program_Chairs · 2023-10-07

**Decision:**

Accept-Findings

**Comment:**

This paper studies the controlled text reduction task. The reviewers find the paper to be well-motivated, and attempt a combination of reasonable strategies. The paper also seems to benefit from good writing. However, the reviewers find the novelty of the work to be extremely limited with most of the techniques borrowed from the literature. The experiment section needs more work, with thorough human evaluation and clearly identifying the source of improvement. Given the importance of the task, this paper could establish baselines for the text reduction task in the future.